# Camera Trapping Reveals Spatiotemporal Partitioning Patterns and Conservation Implications for Two Sympatric Pheasant Species in the Qilian Mountains, Northwestern China

**DOI:** 10.3390/ani12131657

**Published:** 2022-06-28

**Authors:** Dexi Zhang, Bei An, Liuyang Chen, Zhangyun Sun, Ruirui Mao, Changming Zhao, Lixun Zhang

**Affiliations:** 1College of Ecology, Lanzhou University, No. 222, Tianshui South Road, Lanzhou 730000, China; zhangdx19@lzu.edu.cn (D.Z.); chenly19@lzu.edu.cn (L.C.); sunzhy18@lzu.edu.cn (Z.S.); maorr20@lzu.edu.cn (R.M.); zhaochm@lzu.edu.cn (C.Z.); 2Yuzhong Mountain Ecosystems Observation and Research Station, Lanzhou University, Lanzhou 730000, China; anb@lzu.edu.cn; 3School of Basic Medicine Sciences, Lanzhou University, Lanzhou 730000, China

**Keywords:** *Crossoptilon auritum*, *Ithaginis cruentus*, galliformes, activity pattern, species distribution model (SDM), habitat overlap, camera traps

## Abstract

**Simple Summary:**

Camera-trapping technology has been widely applied to obtain survey data and enhance understanding of animal ecology. Ground-dwelling pheasants with limited distributions and weak dispersal capacity are prone to extinction due to disturbances and climate change in high-altitude mountain areas. The Qilian Mountains form a global biodiversity hotspot for endemic species and contain crucial areas for ecological and biodiversity conservation. The Blue Eared Pheasant (EP) and Blood Pheasant (BP) are indicator species of the environment and currently occur in the Qilian Mountain National Nature Reserve (QMNNR). Understanding their stable coexistence is key for making informed conservation and management actions. They have similar daily activity patterns but their monthly activity patterns are strikingly different. Both BP and EP prefer forest habitats but BP nests in more dense vegetation cover. Ninety-one percent of BP distribution falls within EP distribution in the QMNNR. Their areas of potential overlap are in the central and eastern parts of the QMNNR, but landscape connectivity is relatively poor. This study further improved the understanding of the basic knowledge of BP and EP coexistence. Conservation actions should give priority to those highly overlapping areas and strengthen forest landscape connectivity, as they provide irreplaceable habitats for threatened Galliformes.

**Abstract:**

Studying the spatio-temporal niche partitioning among closely related sympatric species is essential for understanding their stable coexistence in animal communities. However, consideration of niche partitioning across multiple ecological dimensions is still poor for many sympatric pheasant species. Here, we studied temporal activity patterns and spatial distributions of the Blue Eared Pheasant (EP, *Crossoptilon auritum*) and Blood Pheasant (BP, *Ithaginis cruentus*) in the Qilian Mountains National Nature Reserve (QMNNR), Northwestern China, using 137 camera traps from August 2017 to August 2020. Kernel density estimation was applied to analyze diel activity patterns, and the Maxent model was applied to evaluate their suitable distributions and underlying habitat preferences. Eight Galliformes species were captured in 678 detection records with 485 records of EP and 106 records of BP over a total of 39,206 camera days. Their monthly activity frequencies demonstrate temporal partitioning but their diel activity patterns do not. Furthermore, 90.78% of BP distribution (2867.99 km^2^) overlaps with the distribution of EP (4355.86 km^2^) in the QMNNR. However, BP manifests a high dependence on forest habitats and shows larger Normalized Difference Vegetation Index (NDVI) values, while EP showed obvious avoidance of forest with NDVI greater than 0.75. Hence, differentiation in monthly activity patterns and partitioning in habitat preference might facilitate their coexistence in spatiotemporal dimensions. Conservation actions should give priority to highly overlapping areas in the center and east of the QMNNR and should strengthen forest landscape connectivity, as they provide irreplaceable habitats for these threatened and endemic Galliformes.

## 1. Introduction

Interspecific interaction is a classical and lively research topic [1,2,3] that studies how ecologically similar species in animal communities partition their niches across multiple dimensions [4,5,6]. Competition between sympatric species (completely or partially overlapping in the spatial distribution of ecologically similar species) has set key mutual constraints by influencing their ability to occupy and exploit limited resources, leading to niche differentiation [7]. To minimize the negative impacts of interspecific competition, species generally tend to be partitioned in time, space, or food resources [6,8,9,10,11]. Animals use time as a resource that can mitigate interference or competition by shifting their activity patterns over the day and month [4,12]. For example, sympatric species of mammals in various orders (e.g., Carnivora, Cetartiodactyla or Rodentia) exhibit significant activity peak dislocations that alleviate competition [3,13,14,15]. Conversely, some species can gain potential benefits (e.g., anti-predator responses) by eavesdropping on signals from key information producers in an animal community [16]. This might facilitate the co-occurrence of mixed-species groups or increase the temporal overlap between sympatric species [16], resulting in niche partitioning in other dimensions rather than a strict demarcation of time [6,17,18]. For spatial dimensions, animals can coexist by dividing their habitat requirements both horizontally and vertically [13,19]. Furthermore, areas where multiple species coexist are of high conservation priority because they provide core habitats for species assemblages [20,21,22]. In addition to interspecific interactions, abiotic factors also play a key role in spatial niche differentiation [23,24]. For instance, Normalized Difference Vegetation Index (NDVI), an indicator of vegetation cover [25], is considered to be one of the most critical factors determining the spatial distribution of Satyr Tragopan (*Tragopan satyra*) [24]. Hence, an in-depth understanding of niche partitioning in spatiotemporal dimensions is not only essential for understanding the stable coexistence of similar species [2,10,26] but also can greatly facilitate the regional conservation management of species at a practical level [27].

As an indicator of the need for ecological and environmental protection [19,28,29], 26% of the Galliformes species are listed as threatened with global extinction [30] and under threat from climate change [31], and endemic species with small areas of distribution at higher elevations are particularly vulnerable [32,33]. The populations of many Galliformes have also faced habitat fragmentation and deterioration caused by human activities, such as the conversion of forests to anthropogenic land and deforestation in the past few decades [30,34,35], which may lead Galliformes to the verge of extinction [36]. Moreover, this group may face more intense interspecific competition than some animals in the wild due to their large size and weak dispersal capacity [6]. In the past decade, the extensive use of camera traps, due to their continuous monitoring advantages, non-invasive effects and limited interference with the natural behavior of wildlife [37,38], has provided unprecedented opportunities to quantify animals’ natural activity through unbiased time-stamped photographs [4,39,40]. Furthermore, quantifying overlapping distributions in large-scale landscapes can help us to better understand how environmental factors (bioclimate, vegetation, topography, human disturbance, etc.) shape the spatial niches of species. In this context, Maxent (maximum entropy modeling), based on the correlation between species occurrence points and environmental factors [41], is a powerful tool to analyze multi-species spatial distribution overlap due to its predictive strength [42,43].

The Qilian Mountains National Nature Reserve (hereafter, the QMNNR) in northern China, known as a center for endemic species and a crucial area for ecological and biodiversity conservation [22,44], harbors 10 species of Galliformes [45]. The Blue Eared Pheasant (*Crossoptilon auritum*, hereafter EP) is endemic to China and tends to occur at higher elevations ranging from 2700 m to 3500 m. Blood Pheasant (*Ithaginis cruentus,* hereafter BP) has a broader distribution beyond China and prefers lower elevations from 1700 m to 3200 m [45]; meanwhile, their distributions are shrinking [46] and their occurrence in the field is rare [47]. These two species are similarly distributed in most ranges of montane forests in the QMNNR [17], and their coexistence may be facilitated by partitioning resources across food [17,48] and foraging strategy [6], time [49], or space [18,27,50]. However, little information is known about their spatio-temporal niche partitioning. Thus, further field studies to understand their habitat use on a mountain-wide scale, particularly the areas of overlap between them, are essential for strengthening habitat protection and management [21].

Here, we conducted monitoring of EP and BP by camera traps in the QMNNR from August 2017 to August 2020, and focused on three key questions: (1) To what extent has partitioning in temporal activity patterns occurred between the two species? (2) Where are the priority conservation areas for both species, identifying their overlapping suitable habitats in the QMNNR? (3) How do they differ in underlying habitat preferences and response to environmental variables? Furthermore, we hypothesized that (1) they would show limited partitioning in activity patterns and spatial distribution due to the co-occurrence of mixed flocks and differences in food composition [17] and that (2) they would exhibit varying degrees of differentiation in their preferences for suitable habitat based on the competitive exclusion principle [2,26].

## 2. Materials and Methods

### 2.1. Study Areas

The QMNNR covering 26,806 km^2^ is a transition area between the arid and semi-arid areas (97°25′–103°46′ E, 36°43′–39°36′ N) and the alpine zone of Qinghai Tibet in Northwestern China (Figure 1) [51,52], which is known as one of the six national “hotspots” for avian biodiversity conservation [20,42] and endemic species centers in China [44]. The vegetation here is composed of vast alpine meadows and shrublands with isolated areas of subalpine coniferous forest [53] composed of Qinghai spruce (*Picea crassifolia*) preferentially distributed on shaded and semi-shaded slopes at 2500–3300 m a.s.l. [54] and Qilian juniper (*Juniperus przewalskii*) that is widely distributed on infertile sunny slopes between 2600 and 4300 m a.s.l. in the QMNNR [55]. Rainfall is mainly concentrated from May to September, with annual precipitation of 200~500 mm and decreases from east to west; the mean temperature is 4.0 °C [51].

### 2.2. Camera Traps Survey

We collected field data from August 2017 to August 2020 in the QMNNR. A total of 137 camera traps (Ltl 6210 and Ltl 6511, Zhuhai, China; EREAGLE E1B, Shenzhen, China; Seagull-LY-1, Shanghai, China) were deployed at six study areas with elevations ranging from 2400 m to 3800 m a.s.l. (Figure 1 and Appendix A). A major landscape type was evergreen coniferous forest dominated by *P. crassifolia* and *J. przewalskii* (at Sidalong, Xiyinghe, and Haxi; hereafter, SDL, XYH, and HX) (Figure 1). To understand their spatial distributions, both horizontal and vertical gradients were considered including other major landscape types, scrub-grassland at low elevation (2783 ± 201.21 m a.s.l., at Qifeng; hereafter, QF) and at high elevation (3353 ± 167.40 m a.s.l., at Longchanghe and Machang; hereafter, LCH and MC) (Figure 1). All study areas were divided into 1 × 1 km grid by the Geographic Information System (ArcGIS 10.2). We randomly selected open and front-lit points to deploy the camera traps in each grid, to reduce the false trigger effect [37]. Camera traps were placed 0.3–1.0 m above the ground, kept active 24 h/day, and programmed to take 3 photographs and 1 video (0.5 min) with a delay of at least 1 min between consecutive events. Depleted batteries and SD cards were replaced and collected approximately every four months. The morphology of the study species makes it impossible to visually distinguish individuals reliably. Therefore, detection records were analyzed with widely accepted methods: (i) one or more individuals of the same species presenting consecutive records taken within 30 min or (ii) consecutive records of individuals of different species [12,38,56,57]. For each detection record, we recorded the site, species, date, time, elevation, and group size (the maximum number of individuals recorded during a 30 min period for a single occurrence event).

### 2.3. Species Distribution Model Construction

#### 2.3.1. Environmental Variables Acquisition and Pre-Processing

To assess the suitable habitats of EP and BP, we used the MaxENT model [41] owing to its desirable statistical properties [18,42,43] with regard to species occurrence data and environmental variables to predict their potential habitat in large-scale spaces [41]. We scored a total of 28 environmental variables (bioclimate, vegetation, topography, and human disturbance: for details see (Appendix A) that were potentially important for EP and BP considering their ecological significance [17,58,59,60]. The 19 bioclimatic factors with a spatial resolution of 30 arc-seconds (1 space km) were downloaded from the climate data on WorldClim (www.worldclim.org, accessed on 11 May 2021) [61] along with the three distance variables, which were generated from the Euclidean distance in ArcGIS 10.2 using the vectors of rivers, roads, and residential points, respectively (Appendix A). All environmental variables were resampled, and the raster layer with a uniform resolution of 1 km was converted into “.asc” format by ArcGIS 10.2, which is required by the MaxENT software [41]. To avoid overfitting caused by spatial autocorrelation and to reduce sampling bias, we removed the repeated occurrences within a 1 km × 1 km cell using the “Wallace” package https://wallaceecomod.github.io/ (accessed on 19 May 2021) [62] based on a maximum of one occurrence point at each grid cell [18,63], since all study areas practically covered the possible habitat range of both species and met the fine resolution of 1 km range. Finally, 47 occurrences of EP and 14 occurrences of BP were used for habitat suitability prediction of the two species.

#### 2.3.2. Modeling Species Distribution

MaxENT 3.4.1 (http://www.cs.princeton.edu/~schapire/maxent/ accessed on 17 May 2021) [41] based on presence-only data [64] was implemented to predict the potential habitat suitability of both species. The program has been widely applied to discriminate between observed presences and absences [21,41,42]. We used 10 bootstrap replicates and randomly split the presence records into training and testing data (75% and 25%, respectively). A pair of variables with Pearson’s correlation > |0.80| was removed and remained the most important factor for consideration in the final model according to permutation importance and ecological meaning for both species [18,65]. Finally, a total of 13 and 12 predictive factors were used to generate two model results, for EP and BP respectively. Jackknife test and logistic output were selected to comprehensively evaluate the importance of environmental factors and represent the logistic values ranging from 0 (lowest probability) to 1 (highest probability) for the distribution of a potentially suitable habitat.

#### 2.3.3. Model Evaluation and Potential Suitable Habitat Classification

The area under the curve (AUC) of receiver operating characteristics [66] was used to measure model performance. Within the possible range 0~1, a larger AUC value indicates a better prediction effect of the model [42] and has random probability when it equals 0.5 [43]. The model evaluation reliability is high when the AUC value exceeds 0.9 [64]. We converted our results to presence and absence predictions based on the threshold values that maximized training sensitivity plus specificity for EP and BP to determine the division of potentially suitable habitats by a binary transformation [67,68]. Afterwards, this threshold value was used to classify suitable (≥threshold value) or unsuitable habitats (<threshold value) for each species.

### 2.4. Data Analyses

To assess monthly activity patterns of EP and BP, we calculated the percentage ratio of detection records for each month (the monthly number of detection records for each species divided by total records for both species per month), to decrease the difference between comparisons of absolute numbers of events. This was a proven method for reliable estimation of monthly activity levels, since the number of camera traps and monitoring days for both species was equivalent throughout the study period [12]. Similarly, we also assessed a monthly change in the number of different group sizes captured for each month in which camera traps were active. Based on the breeding phenology of the two target species [17,58] and local climate characteristics in the QMNNR [55,69], we divided the year into breeding (from April to July), non-breeding (from August to October) and winter seasons (from November to March). To explore if the degree of interspecific overlap in activity between EP and BP changes across seasons, we estimated the diel activity for both species by fitting a non-parametric circular kernel density estimation function [39] to the radian-transformed occurrence records, regarded as a random sampling from 24 h per day reflecting the animals’ maximum true diel activity pattern [39]. We implemented and visualized with the “overlap” package [70] and used the coefficient of overlap (∆_4_ or ∆_1_) to estimate temporal niche differentiation between both species, whereby ∆ can range from 0 (no overlap) to 1 (complete overlap) [39,71]. When the small samples (detection records) were less than 75, ∆_4_ was used as the parameter estimation; otherwise, ∆_1_ was used [70]. Then, we computed the 95% confidence intervals (hereafter, 95% CIs) for more robust overlap estimates by using 10,000 bootstrap replicates [40]. To test whether there were differences in the kernel density curves between the two species across seasons, the Wald test with the “activity” package [72] was used to obtain the test value (significance level at 0.05) by nonparametric bootstrap resampling with 10,000 iterations [40]. All analyses were performed in R version 4.0.5 [73].

To map the horizontal overlap in the habitat between EP and BP, the reclassified habitat maps for each species were overlaid to generate the final vector map of overlapping suitable habitats for both species by using ArcGIS 10.2. We used elevation data from the field detection records of camera traps to pair and compare vertical gradients for EP and BP in the horizontal overlap region (SDL and LCH areas; XYH and HX areas) predicted by the model and in the whole QMNNR. To test whether there was a difference in vertical spatial niche partitioning between the two species, we used the Wilcoxon test method to pair and test because of the non-normal distribution of the elevation data (Shapiro–Wilk test, SDL, and LCH areas: W = 0.949, *p* < 0.01; XYH and HX areas: W = 0.928, *p* < 0.01; QMNNR: W = 0.968, *p* < 0.01) performed in R version 4.0.5 [73], and level of significance set at *p* = 0.05 or *p* = 0.01. Furthermore, the first four environmental variables based on model prediction were selected for analysis of suitable habitat preference according to the jackknife test of the importance of the environmental variables.

## 3. Results

### 3.1. Camera Traps Survey Records

Eight Galliformes species (belonging to 1 family, 7 genera) were identified from 678 detection records (Appendix A) including 485 records of EP and 106 records of BP with 39,206 camera-days effort (Appendix A). Among them, the Chinese Grouse (*Tetrastes sewerzowi*) is listed as Near-threatened (NT) by IUCN (2021), and a China nationally protected species (Category I), and the Chestnut-throated Partridge (*Tetraophasis obscurus*) as Category I. Moreover, both *T. sewerzowi* and *T. obscurus* are endemic species and were mainly captured in the center and east QMNNR. The remaining four species were distributed at the center and west QMNNR. Of the six research areas, SDL in the central QMNNR shows the greatest richness area for Galliformes with six species. The EP was photographed at 69 sites in all six sampling areas while BP was captured at 18 sites in SDL, XYH, and HX (Appendix A).

### 3.2. Temporal Activity Patterns

For monthly activity, EP presented a higher activity frequency than BP for all months except September and was most frequently recorded from January to August, whereas BP was mostly captured from September to December (Figure 2a). Group sizes of both species were predominantly one or two individuals from March to June, while larger groups (≥3 individuals) frequently occurred from July to February with a similar trend in both (Appendix A). Kernel density estimation models indicated that the two species were both fully diurnal (Figure 2). There was a high degree of activity rhythm overlap between the two species during the breeding (**∆_1_** = 0.903, 95% CIs = 0.701–0.927, Figure 2b), non-breeding (**∆_1_** = 0.844, 95% CIs = 0.721–0.935, Figure 2c), and winter seasons (∆_1_ = 0.845, 95% CIs = 0.721–0.936, Figure 2d), and their activity patterns were no different in all three seasons (Wald test, *p* > 0.05). We found that their diel activity patterns exhibited approximate bimodal patterns during the breeding and non-breeding season while they were unimodal between 9:00–16:00 in winter. The peak activity of EP was between 8:00–10:00 a.m., whereas the peak of BP was at 14:00–16:00 in the non-breeding season (Figure 2c).

### 3.3. Spatial Distribution Partitioning

The mean ± SD of the AUC value for ten replicate runs of the modeling was 0.972 ± 0.025 and 0.965 ± 0.044 for EP and BP, respectively (Appendix A), indicating that the model could be used to map their overlapping region of spatial distribution. The potential suitable habitat was 4355.86 km^2^ for EP, accounting for 16.25% of the QMNNR and 2867.99 km^2^ for BP accounting for 10.70% of the QMNNR. Overlapping areas of both species were 2603.61 km^2^, accounting for 90.78% of the total suitable habitat of BP and 59.77% of the total suitable habitat of EP. The most highly suitable habitats were mainly in the central and eastern areas of the QMNNR for both species, while there was little suitable habitat for EP in the west (Figure 3a). The results also indicated that their habitats were fragmented, particularly in the areas where suitable habitat overlapped (Figure 3b,c).

The overall vertical distribution based on camera trapping revealed that EP (3046 ± 220 m a.s.l.) occurred at significantly higher elevations than BP (2955 ± 115 m a.s.l.) (Wilcoxon test, *p* < 0.01, Figure 3d) in the QMNNR. The vertical pattern in the center was similar to that of the whole (*p* < 0.01, Figure 3e), but showed no significant difference in the eastern part of the QMNNR (*p* = 0.226, Figure 3f). As predicted for the distribution of the horizontal suitable habitat, EP also had a wider range in vertical distribution.

### 3.4. Habitat Selection Preference

The potential suitable distribution of the EP was mainly correlated with the normalized difference vegetation index (NDVI), precipitation during the coldest quarter (PCQ), altitude, and maximum temperature of the warmest month (MTWM) (Figure 4a). For the BP, the most important environment variables were distance to settlements (DTS), global land cover (GLC), NDVI, and annual mean temperature (AMT) (Figure 4b). We also found that factors of the slope, aspect, and distance to rivers and roads contributed little to habitat suitability for both species (Figure 4).

Response curves revealed the direction of effects of the four most important variables in modeling the distribution of habitat suitable for EP (Figure 5a–d) and BP (Figure 5e–h). We found that the probability of the occurrence of EP showed a distinct decline where the value of NDVI was greater than 0.75 (Figure 5a). Occurrence probability sharply decreased with increasing PCQ (>4.5 mm) (Figure 5b) and happened in mid-elevation ranges (2500–3500 m a.s.l.), and MTWM range from 10 °C to 20 °C (Figure 5c,d). The occurrence probability of BP increased with increasing NDVI (Figure 5g), and the highest tendency towards forest as the category of GLC (Figure 5f). DTS and AMT each exhibited a single peak, with a maximum occurrence probability of occurrence at 60 km from settlements and mean 0 °C, respectively (Figure 5e,h).

## 4. Discussions

Our results provided evidence that differentiation in monthly activity patterns and partitioning in habitat preference might contribute to the coexistence of two pheasant species in the Qilian Mountains. According to the classical competitive exclusion principle [2,26], the long-term stable coexistence of two sympatric species is facilitated by ecological niche differentiation [3,6,12,18,48]. As predicted by MaxENT, suitable habitat selection of BP is highly associated with forest habitats (Figure 5f,g) while EP demonstrated obvious avoidance of the densest forest (Figure 5a). Previous EP studies focused on population-level descriptions of ecology and habits, and information on activity rhythm was limited to direct observation [17,60]. In this study, random sampling for 24 h per day captured the natural diel activity pattern of the species in as unbiased a manner as possible [4,39]. Diel activity patterns of both species were bimodal in breeding and non-breeding seasons, which were generally consistent with previous studies [17,58,74,75]. However, they showed a unimodal pattern in winter. Seasonal variation in the pronounced peak of activity rhythm might be an adaptive strategy to cope with the changes in daylight, temperature, or food resources [10,14,49,76]. This might reflect a need for the two species in the QMNNR to prolong their midday activity in winter to adapt to inclement weather conditions and maximize use of daylight in seeking out scarce food resources.

The diel activity of EP and BP, without significant differences (Figure 2b–d), provided little evidence of temporal partitioning. Contrary to our first prediction, however, EP and BP exhibited significantly different monthly activity patterns (Figure 2a), which has also been reported in Yunnan [74]. This might be a potential mechanism promoting their coexistence during the late breeding period. Luo et al. [48] reported that large BP groups expand their activity ranges after clustering, which may explain why, in this study, more records were obtained (Figure 2a) and big groups occurred (Appendix A) in September. Such monthly activity variations may promote their coexistence and reflect their diversification in temporal niches. The two species’ co-occurrence were captured in single images (Appendix A), which provided direct evidence for their forming mixed groups, compared to descriptive reports [17]. Similar co-occurrence phenomena also occur between other pheasant species [18,48], especially those sympatric species of different genera, such as Snow Partridge (*Lerwa lerwa*) and Tibetan Snowcock (*Tetraogallus tibetanus*) [18]. There are two possible explanations for the co-occurrence of sympatric Galliformes. The differentiation in their diet [17,48] or foraging strategies [6,48] may have minimized potential interspecific competition. On the other hand, co-occurrence may share reliable and communicable information [59], such as eavesdropping on heterospecific alarm signals, which can potentially benefit both species when either one of them detects a predator [16].

Furthermore, BP has a narrower horizontal and vertical spatial niche (Figure 3). The overlap of 90.78% in suitable habitats of BP overlapping with that of EP indicates that the spatial distributions of both species are similar yet the suitable habitat range of BP is smaller. As previously demonstrated, the food choices of BP are much narrower than EP [17], which may also be a basis for its narrower spatial niche. Analogously, this may also be explained by the conclusion that the higher detection records of EP reflect the larger population of the species ([38,56], Appendix A). To our knowledge, there have been few studies of the detailed spatial distributions of an assemblage of pheasant species within the montane landscape [19,36,45,75]. Our study provided additional valuable information about the distribution of six other Galliformes species in the Qilian Mountains, most of them poorly known (Appendix A), although they were not photographed sufficiently often for further analysis. Generally, the habitats of endemic species should be given higher conversation priority than others, due to their acting as keystones for avian diversity [32,44]. Furthermore, mountain forests can provide habitats for both high and middle-elevation mountain breeding birds in winter [77]. However, although the middle and eastern part of the QMNNR is the core distribution area for these two species and for the other two endemic species of Galliformes (*T. sewerzowi* and *T. obscurus*) (Appendix A), the connectivity of the suitable habitat is poor and the fragmentation of the landscape is serious (Figure 3). More worryingly, arid and semiarid forests have been seriously degraded due to overgrazing, showing that they are vulnerable [55,78], tree boundaries have shifted upwards and the death of *J. przewalskii* saplings has increased in the QMNNR [55]. Thus, our results also imply that conservation plans for Galliformes should consider strengthening the connectivity of forest landscapes [56,58] in the QMNNR in the future as they provide potentially stable and suitable habitat particularly for endemic species of ground-dwelling Galliformes [52,77].

NDVI and temperature were the two environmental factors important for both species, according to the model prediction, but their responses to the environmental variables showed differences (Figure 5). For example, we found that EP shows avoidance when NDVI is higher than 0.75, while BP prefers more dense vegetation. The elevational range of the BP was reportedly 4000–4400 m a.s.l. in the Yunnan Province [74], and 3700–4700 m a.s.l. in southeastern Tibet [19]. Our findings indicated that in the QMNNR, BP was found lower (2600–3300 m a.s.l.) than those in previous studies [19,74] but similar to the elevations (2400–3800 m a.s.l.) in Wanglang and Wolong in Sichuan Province [75]. This might be determined by their high forest dependency and the variations in the ranges of forest boundaries among different regions affected by slope, aspect, topography and climate. For example, the lower limit of *J. przewalskii* occurs at 2600 m in the Qilian Mountains [79], whereas its higher distribution range in southeastern Tibet (4200–4700 m a.s.l) [19] suggests a correlation with the higher elevation occurrence of BP there. In addition, our results demonstrated that the higher habitat suitability of EP and BP is strongly influenced by the temperature of the warmest month and annual mean temperature, respectively (Figure 4), which might be explained by the fact that temperature directly affects the distributional and seasonal changes of vegetation and then indirectly affects the potential distribution of Galliformes in the QMNNR [55,69]. There is a significant positive correlation between temperature and NDVI in the vegetation-climate relationship [69]. Like most other terrestrial animals [80,81], Galliformes are also subjected to the widespread negative impacts of climate change across the world, such as shrinking distribution and greater habitat fragmentation [32,33]. Although temperature and precipitation effects on their suitable habitat were detected under the current climate conditions, these might be specific and immediate patterns due to their close relationship with the current tree line. In this regard, long-term field monitoring is critical to assess the community dynamics of Galliformes and predict changes in their future distribution.

In this study, although our field protocol for camera trap layout was not solely to capture data on ground-dwelling Galliformes, the information provided is important given that no systematic study has reported before on the spatial-temporal niche partitioning of the two pheasant species. For the spatial distribution, the small data set of occurrence points, especially for BP due to their low occurrence in the field [47], did not allow our study to analyze variations in seasonal distribution. Therefore, we concede that there might be some inter-seasonal noise that influences spatial niche patterns. In addition, it would be interesting to combine the partitioned time budget of different behaviors (e.g., vigilance, resting, and feeding) with the information on dietary niches [11], such as differentiation of foraging strategies, and feeding preferences [6,48].

## 5. Conclusions

Our study revealed for the first time the extent of overlapping activity patterns and partitioning of suitable habitat of EP and BP in the QMNNR, Northwestern China. The results provide key information for future studies of pheasant species interactions. We identified a high degree of overlap in their diel activity patterns and suitable habitats on a mountain-wide scale but found obvious differences in monthly activity patterns. BP mainly manifested in its high dependence on dense forest habitats with larger NDVI values while EP manifested avoidance of vegetation with NDVI values greater than 0.75. Hence, differentiation in monthly activity patterns and partitioning in habitat preferences might facilitate their coexistence. Furthermore, the use of camera traps in long-term monitoring projects demonstrates the occurrence of several regionally rare endemic species, coexisting with BP and EP, and provides information for interspecific conservation strategies. To maintain and protect biodiversity in the future, we suggest that conservation actions should be directed at strengthening the protection of suitable habitat networks, especially in the middle and eastern parts of the QMNNR. Finally, this study highlights the significant advantages of camera trap technology for modeling the interactions between these sympatric terrestrial birds and informing conservation plans by identifying their core suitable habitats.

## Figures and Tables

**Figure 1 animals-12-01657-f001:**
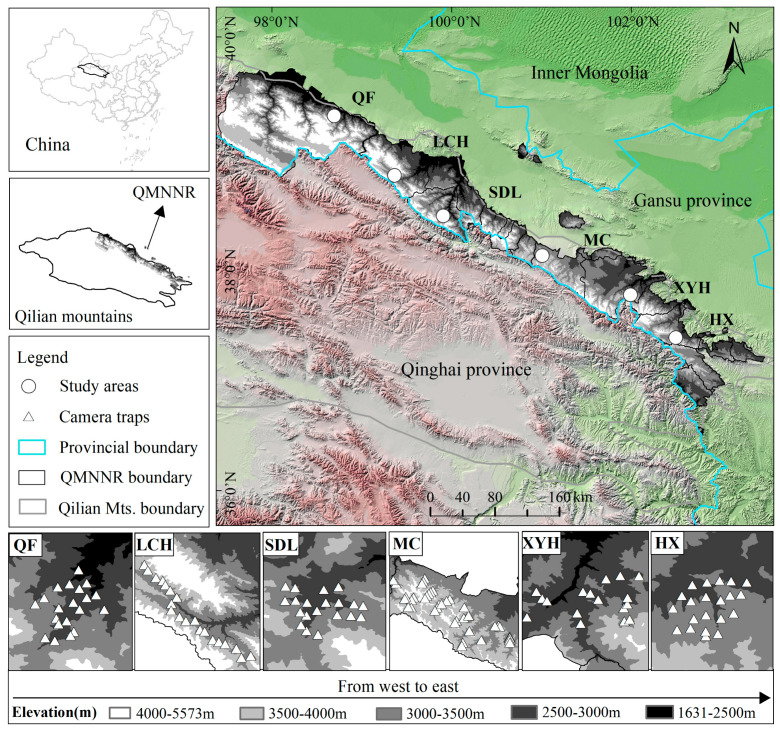
Camera–trap site distributions in six study areas (Qifeng, QF; Longchanghe, LCH; Sidalong, SDL; Machang, MC; Xiyinghe, XYH, and Haxi, HX) in the QMNNR, Northwestern China. (Available online for the map layer of Altitude and Remote Sensing Image: www.gscloud.cn/ accessed on 12 April 2021).

**Figure 2 animals-12-01657-f002:**
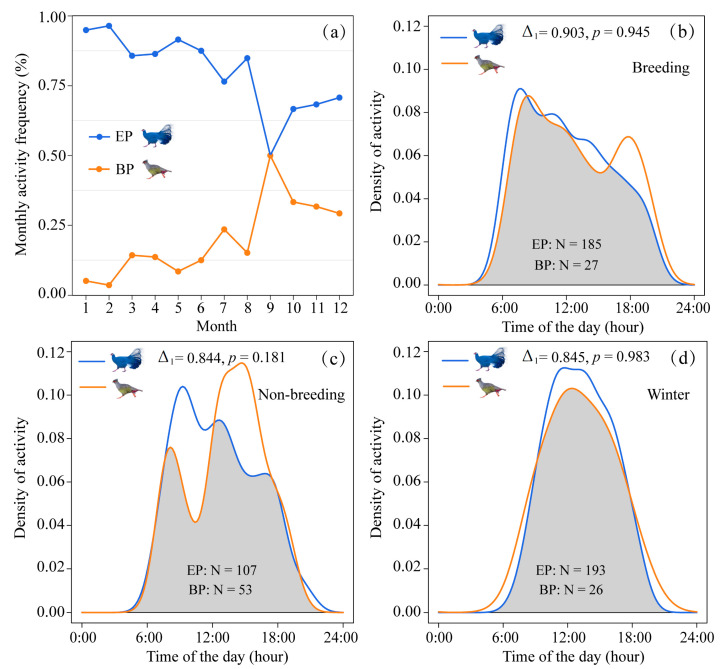
Temporal activity patterns of EP and BP in the QMNNR of Northwestern China. (**a**) Monthly activity patterns expressed as percent frequency. Diel activity and overlap of EP and BP in (**b**) breeding, (**c**) non-breeding, and (**d**) winter season. Gray shading shows the activity overlap of the two species, and the number at the top center of each graph represents the mean coefficient of overlap and the significant difference used in the Wald test.

**Figure 3 animals-12-01657-f003:**
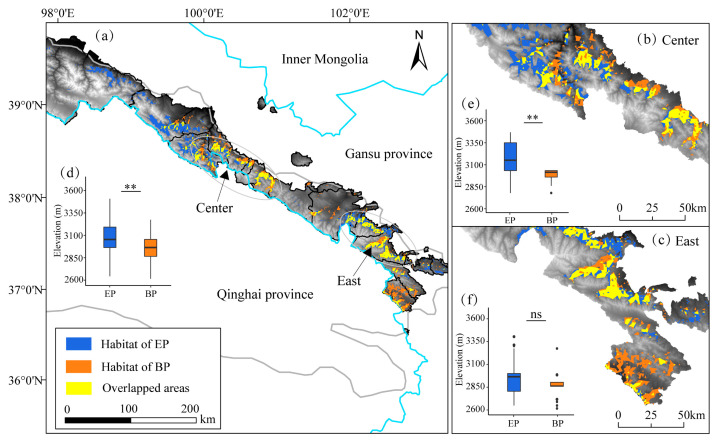
The spatial distribution pattern of EP and BP in the QMNNR, Northwestern China. (**a**) Potentially suitable habitat and overlap of both species in horizontal space are shown by modeling. (**b**,**c**) Higher resolution maps of concentrated areas of overlap in the middle and eastern sections in the QMNNR, as shown in (**a**,**d**–**f**) indicate their vertical spatial distribution in the QMNNR, Middle and East of the QMNNR based on the records by camera traps, respectively. The double star indicates that the differences were significant at the *p* = 0.01 level based on the Wilcoxon test, but showed no significant difference in the eastern part of the QMNNR (*p* = 0.226, Figure 3f).

**Figure 4 animals-12-01657-f004:**
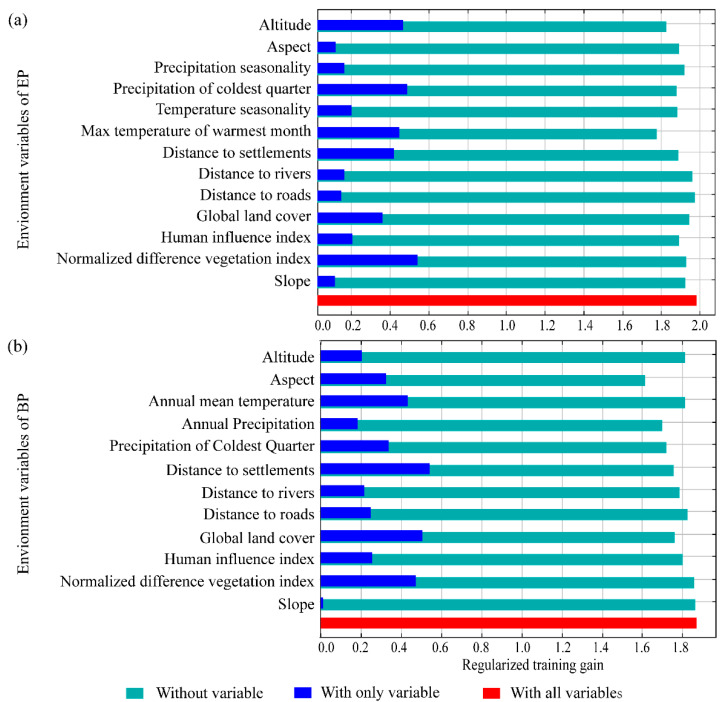
Jackknife test of the importance ranking of environmental variables by the habitat suitability models for (**a**) EP and (**b**) BP. The dark blue band represents the gain to the prediction result when each variable of the model is run separately. The light blue band shows the gain of the model when removing this factor and the model is run only with other variables. The red band at the bottom indicates the gain results when the model uses all variables. The longer the dark blue band, the more important the predictor variable.

**Figure 5 animals-12-01657-f005:**
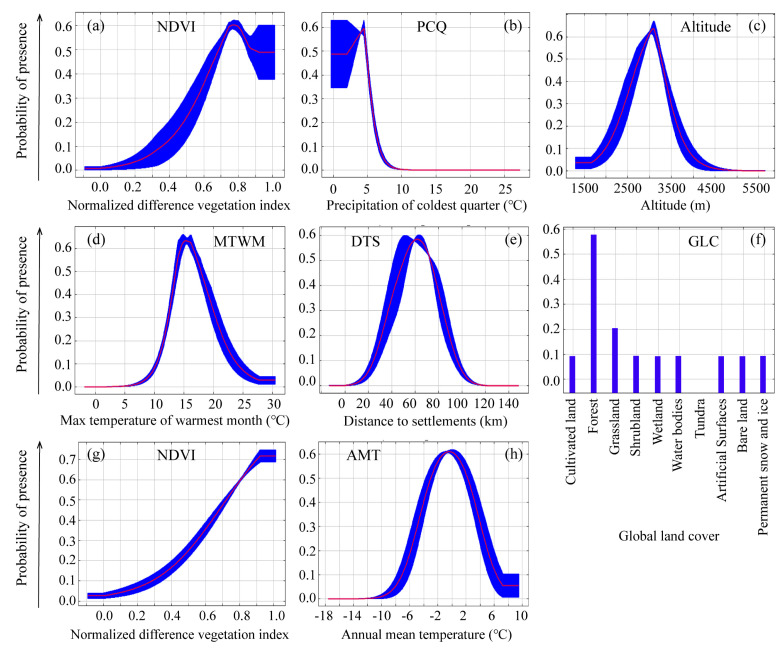
Response curve results of MaxENT modeling of the first four environmental variables for potentially suitable habitat for EP (**a**−**d**) and BP (**e**–**h**). These response curves were generated for the most important variables (the top four in each model) and show the mean response of the cross-validated models with 10 replicate runs (red line) and mean ± one standard deviation (blue band).

## Data Availability

The datasets used in this study are available from the corresponding author on reasonable request.

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
