# Peer review of "Camera Trapping Reveals Spatiotemporal Partitioning Patterns and Conservation Implications for Two Sympatric Pheasant Species in the Qilian Mountains, Northwestern China"

_animals, 2022, doi:10.3390/ani12131657_

Round 1
Reviewer 1 Report
This is a study of niche partitioning between two sympatric pheasant species in the Qilian Mountains National Nature Reserve (QMNNR), northwest China, from August 2017 to August 2020. The authors studied temporal activity patterns and spatial distributions of Blue Eared Pheasant (Crossoptilon auritum) and Blood Pheasant (Ithaginis cruentus) using camera traps. They investigated (1) whether temporal activity patterns differed between the species, (2) whether their core habitats overlapped, and (3) identified key environmental variables affecting species’ distributions. They set 137 infrared camera-traps in six research areas, at altitudes of 2400m-3800m. In total, 678 photographs of pheasants were taken during 39,206 camera-days, that included 485 records of C. auritum and 106 records of I. cruentus. The findings showed a high overlap of diel activity patterns and habitat distribution with divergences in monthly activity patterns and habitat preferences resulting in niche differentiation. This study provides insights into the two species’ abundance, daily activity patterns and potential habitats by identifying key ecological factors in their mountain ecosystems and highlighting the importance of forest landscape connectivity for pheasant conservation.
It is obvious that a lot of work went into this study and there are many interesting Figures and Tables. Moreover, it is of high relevance to conservation, as it focuses on the Qilian Mountains National Nature Reserve (QMNNR), one of the six national “hotspots” for avian biodiversity conservation and endemic species centers in China. Also, one of the pheasant species studied (C. auritum) is endemic.
Nevertheless, the text needs a lot more clarity as it is not easy to follow the flow of ideas, and confusing to understand at some points. The manuscript, including legends of Figures and Tables, should be corrected by a native English-speaker as there are many language errors and extensive editing of the English language and style are required. Additionally, the manuscript needs to be re-written with clearer and more comprehensive information regarding its relevance for the conservation of the two pheasant species.
Reviewer 2 Report
A useful paper that uses a novel method to look at the distribution and habitat preferences of two species of endangered pheasant.
The simple summary is very detailed and technical. A non-science reader is going to struggle with some of the words and phrases. For example, montane, galliformes, sympatric. Perhaps use common names rather than binomially names, or speak generally about the pheasants in an inclusive manner.
In the simple summary you state that montane-dwelling Galliformes are some of the most vital groups of avifauna. But you do not say why this. What makes them vital?
I would recommend that you use abbreviations for the common name of each species throughout the paper. It would make readability much improved and accessible. So instead of using the scientific name perhaps use EP for blue-eared pheasant and BP for blood pheasant.
Abstract
The conservation applications in the abstract are very useful. Are they mirrored in the discussion? I would include further evaluation of the conservation relevance of your work in the discussion.
introduction
Line 50 remove the "do" so it is "that studies how ecologically..." If you keep in "how do..." you turn the sentence into a question and therefore you will need a question mark.
Recommend that you define what you mean by sympatric early on in the introduction.
Line 89 does Maxent need a citation?
Recommend that you include the IUCN Red List Assessment status for the blood pheasant and blue-eared pheasant and include any information on population trends (i.e. give some conservation evidence for why this project is needed).
Should it be the Qilian Mountains?
Line 112 to 114 can you provide some justification for your hypotheses based on the information that you have reviewed in your introduction?
Methods
Line 134: The scientific name of the Qilian juniper seems to change from P. to S. Please confirm.
Line 144: do you mean memory cards (SD cards) when you say memory records?
Figure 1 is very helpful. Does it require a citation or permission for use?
Line 180 what correlated variables were removed?
Line 183 can you expand on the predictor variables?
Line 231 non-normal? Rather than abnormal distribution.
Line 234 which tests used the more conservative level of significance of 0.01?
Figure 2 how was monthly activity frequency calculated? Please can this be explained either in the data analysis section or under the graph.
What behaviours are included in activity?
What is a density of activity? The Y axis on this graph needs units. Again, how was this calculated?
Figure 2 is potentially useful but currently the reader is unable to judge what it shows because these data that have been used to build the figure cannot be identified.
Please explain Figure 4. What do you mean by without variable, with one variable and with all variables?
What are these variables? They need to be explained to the reader.
What are the units for Figure 4?
Figure 5 please check your spelling for X axes on this figure as there are multiple spelling mistakes (e.g. normalised is spelled incorrectly, as is vegetation).
You need to mark on the Global Land Cover graph which bar represents which type of terrain.
Discussion
Line 421 "the information provided"
Line 431 should their behaviour patterns be considered too? I.e. when they are likely to be active and their social system? As these will influence the types of behaviours performed.
Line 451 should this be in a spatiotemporal dimension? Or should dimensions be plural?
There is no ethical review statement provided in this paper. Please provide information on how the methods were scrutinised for scientific integrity.
Reviewer 3 Report
Manuscript ID animals-1737621. “Camera Trapping Reveal Spatiotemporal Partitioning Patterns 2 and Conservation Implications of Two Sympatric Pheasant 3 Species in Qilian Mountains, Northwestern China”.
The authors describe a study of the spatiotemporal niche partitioning of two Pheasants; Blue Eared Pheasant (Crossoptilon auritum) and Blood Pheasant (Ithaginis cruentus), in a National Nature Reserve in the Northwest China. Activity of pheasants is recorded by 137 camara traps from August 2017 to 2020. Kernel density estimation was applied to analyze diel activity patterns and Maxent model to evaluate the spatial distribution of the two species and habitat preferences. There was at high degree of overlap in diel activity patterns between the two species and little evidence was found for their diel activity partitioning despite in monthly activity patterns. Blood Pheasant preferred forest habitat with high NDVI values, while Blue Eared Pheasant preferred habitats with low NDVI values.
- General concept comments
Method and results are well written although there is no introduction to the NDVI index. The abstract, introduction and discussion need to be rewritten without unnecessary words, filler words, paragraphs with no exact information and boast. The stile is unscientific. Figures are very nice and in fine resolution.
Abstract, introduction and discussion should be changed to a shorter and more precise language.
Examples of unnecessary words, paragraphs without exact information and boast.
Line 14: one of the most vital groups of avifauna over the worldwide
Line 24: These results provided valuable insights
Line 50: Interspecific interaction is an old but lively research axis
Line 60: Nevertheless, in an animal community, some species are “keystone” information producers while others are seekers of information
Line 76: As a barometer of ecological and environmental protection
Etc. many sentences contain a lot of fill in words like mechanism, nevertheless, certainly, some way, a lot of, interestingly etc.
- The manuscript is clear, and relevant for the field, but it needs to be rewritten in a more scientific humble manner
- There are few publications referred to within the last 5 years, however, this may be due to the subject and lack of knowledge of pheasant habitats
- The experiment and design is scientific sound and appropriate to test the hypothesis
- Results seems reproducible based on the details given in the methods section
- The figures/tables/images are fine and they are easy to interpret. However, figure five seems confusing. The two species are not examined pairwise by the same parametres. and understand? Is the data interpreted appropriately and consistently throughout the manuscript? Please include details regarding the statistical analysis or data acquired from specific databases.
- The conclusions consistent with the evidence and arguments presented
- There are no ethical statements, however camera traps are a noninvasive monitoring method, no animals are harmed.
Round 2
Reviewer 1 Report
1. In some parts of the text, elevation in thousands is written with a comma (e.g. lines 101 & 102) and in other parts there is no comma (e.g. lines 128 & 129). This should be corrected throughout the manuscript, including the Figure legends (e.g. Figure 1)
2. It should be ‘the QMNNR’ throughout the manuscript (e.g. Line 290)
3. ‘northwest’ should be replaced with ‘Northwestern’ throughout the manuscript, including Figure legends (e.g. Figure 3)
4. Line 59: should be plural, ‘patterns’
5. Lines 67-69: improve the phrasing ‘Furthermore, areas where multiple species
coexist are of high conservation priority because they provide core habitats for species assemblages’.
6. Some parts of the text mention ‘Interspecific’ (e.g. Line 51) and others ‘inter-specific’ (e.g. line 69). Please check and make sure to use the same format throughout the manuscript.
7. Line 78: should be rephrased ‘26% of the Galliformes species are listed …’
8. Line 87: delete ‘very’
9. Line 150: delete ‘consistent’
10. Line 152: replace ‘for different species’ with ‘of different species’
11. Line 154: should be ‘a 30 min period’
12. Line 168: should be ‘1 space km’
13. Line 173: should be ‘the MaxENT software’
14. Line 200: replace with ‘Afterwards’
15. Line 247: should be ‘and a China nationally protected species ….’
16. Line 248: should be ‘and the Chestnut-throated Partridge’
17. Line 248: delete ‘is listed’
18. Line 250: should be ‘were distributed’
19. Line 273: should be ‘at the top center’
20. Line 274: should be ‘in the Wald test’
21. Line 280: delete ‘respectively’
22. Line 291: ‘has’ should be ‘had’
23. Line 297: replace ‘indicated’ with ‘indicate’
24. Line 332: replace ‘month’ with ‘monthly’
25. Line 334: delete ‘in the spatiotemporal dimensions’
26. Line 336: delete ‘certainly’
27. Line 365: replace ‘potentially’ with ‘potential’
28. Line 529: the scientific name should be in italics
Reviewer 2 Report
I am happy with the thorough edits that the reviewers have completed and I am pleased to see extra assistance has been sought with the preparation of the manuscript and the written English. This is a much improved and useful paper and I am happy to recommend for publication.
Author Response
Once again, we gratefully appreciate your valuable comments and suggestions on the Review Report (Round 1) for our manuscript, as well as the essential guide to our future research.